# Association of Computed Tomography Scan-Assessed Body Composition with Immune and PI3K/AKT Pathway Proteins in Distinct Breast Cancer Tumor Components

**DOI:** 10.3390/ijms252413428

**Published:** 2024-12-14

**Authors:** Ting-Yuan David Cheng, Dongtao Ann Fu, Sara M. Falzarano, Runzhi Zhang, Susmita Datta, Weizhou Zhang, Angela R. Omilian, Livingstone Aduse-Poku, Jiang Bian, Jerome Irianto, Jaya Ruth Asirvatham, Martha Campbell-Thompson

**Affiliations:** 1Division of Cancer Prevention and Control, Department of Internal Medicine, College of Medicine, The Ohio State University, Columbus, OH 43201, USA; 2Department of Pathology, Immunology and Laboratory Medicine, College of Medicine, University of Florida, Gainesville, FL 32611, USA; fudo@pathology.ufl.edu (D.A.F.); sfalzarano@ufl.edu (S.M.F.); zhangw@ufl.edu (W.Z.); thompmc@pathology.ufl.edu (M.C.-T.); 3Department of Biostatistics, College of Public Health & Health Professions & College of Medicine, University of Florida, Gainesville, FL 32611, USA; zhangrunzhi886@gmail.com (R.Z.); susmita.datta@ufl.edu (S.D.); 4Department of Cancer Prevention and Control, Roswell Park Comprehensive Cancer Center, Buffalo, NY 14203, USA; angela.omilian@roswellpark.org; 5Department of Epidemiology, College of Public Health & Health Professions & College of Medicine, University of Florida, Gainesville, FL 32611, USA; adusepokul@ufl.edu; 6Department of Health Outcomes & Biomedical Informatics, College of Medicine, University of Florida, Gainesville, FL 32611, USA; bianjiang@ufl.edu; 7Department of Biomedical Sciences, College of Medicine, Florida State University, Tallahassee, FL 32306, USA; jerome.irianto@med.fsu.edu; 8Department of Pathology, Baylor, Scott & White Health, Temple, TX 76502, USA; ruth.asirvatham@bswhealth.org

**Keywords:** skeletal muscle, adipose tissue, breast cancer, tumor immune microenvironment, phosphoinositide 3-kinases

## Abstract

This hypothesis-generating study aims to examine the extent to which computed tomography-assessed body composition phenotypes are associated with immune and phosphoinositide 3-kinase (PI3K)/protein kinase B (AKT) signaling pathways in breast tumors. A total of 52 patients with newly diagnosed breast cancer were classified into four body composition types: adequate (lowest two tertiles of total adipose tissue [TAT]) and highest two tertiles of total skeletal muscle [TSM] areas); high adiposity (highest tertile of TAT and highest two tertiles of TSM); low muscle (lowest tertile of TSM and lowest two tertiles of TAT); and high adiposity with low muscle (highest tertile of TAT and lowest tertile of TSM). Immune and PI3K/AKT pathway proteins were profiled in tumor epithelium and the leukocyte-enriched stromal microenvironment using GeoMx (NanoString). Linear mixed models were used to compare log2-transformed protein levels. Compared with the normal type, the low muscle type was associated with higher expression of INPP4B (log2-fold change = 1.14, *p* = 0.0003, false discovery rate = 0.028). Other significant associations included low muscle type with increased CTLA4 and decreased pan-AKT expression in tumor epithelium, and high adiposity with increased CD3, CD8, CD20, and CD45RO expression in stroma (*p* < 0.05; false discovery rate > 0.2). With confirmation, body composition can be associated with signaling pathways in distinct components of breast tumors, highlighting the potential utility of body composition in informing tumor biology and therapy efficacies.

## 1. Introduction

Understanding how body composition affects tumor markers and patient prognosis is critical for precision medicine in breast cancer. High total adipose tissue (TAT) mass and low skeletal muscle tissue mass, i.e., sarcopenia, are independently associated with overall survival among patients with nonmetastatic breast cancer [1]. This risk classification using adipose and skeletal muscle, which can be assessed by computed tomography (CT) or other imaging tools, is more precise than body mass index (BMI) for predicting mortality [1]. The mechanisms between body composition components and mortality can be multiple. For example, patients with (vs. without) sarcopenia are at higher risk of treatment-related toxicity [2]. Another potential mechanism is that body composition may be related to tumor signaling pathways that are associated with patient outcomes [3]. However, current evidence is very limited for patients with breast cancer. Understanding the implications of body composition on tumor signaling pathways is clinically important because this knowledge can be integrated to better inform tumor biology and treatment efficacies. 

Among tumor signaling pathways, both immune and phosphoinositide 3-kinase (PI3K)/protein kinase B (AKT) pathways can be impacted by adipose and skeletal muscle tissue. Obesity (BMI ≥ 30 kg/m^2^) adversely affects the breast tumor immune microenvironment by reducing stromal tumor-infiltrating lymphocytes (TILs) and increasing immunosuppression [4,5]. Interestingly, data from patients with melanoma, kidney cancer, and lung cancer who receive immune checkpoint inhibitors (ICIs), including programmed cell death protein 1 (PD-1) or cytotoxic T-lymphocyte–associated protein 4 (CTLA4) checkpoint inhibitors, show that those with obesity or overweight have a better overall and progression-free survival than those without obesity [6,7,8]. Skeletal muscle enhances the production of natural killer cells through the secretion of myokines [9], but it is unclear whether muscle plays a role in influencing other immune cell types and tumor immune microenvironment. In addition, obesity is linked to the activation of the PI3K/AKT pathway, including mechanistic target of rapamycin kinase (MTOR) and other downstream proteins [10,11,12,13,14]. However, MTOR is also a key factor in muscle synthesis [15]. Patients with breast cancer and with higher (vs. lower) physical activity levels have higher MTOR activities in tumors [16]. To what degree PI3K/AKT/MTOR pathway activation in breast tumors is associated with skeletal muscle as a component of body composition is unclear. 

The objective of this study was to explore the extent to which body composition components are associated with immune and PI3K/AKT signaling pathways in breast cancer. We classified women with breast cancer into four mutually exclusive body composition types—adequate, high adiposity, low muscle, and high adiposity with low muscle—and correlated these types with protein expression levels in breast tumors. A multiplex, spatially-resolved method was used to measure protein expression in the epithelium and stroma components of tumors. We also compared these associations with those between conventional anthropometry measurements, i.e., BMI and waist circumference (WC). The associations identified in this study would shed new light on studying body composition and these pathway markers as well as patient outcomes in large confirmatory studies. 

## 2. Results 

Table 1 gives the patient demographic and clinical characteristics. The mean (standard deviation [SD]) age at breast cancer diagnosis was 56.8 (10.5) years. In total, 82.7% (*n* = 43) of individuals identified as being of White race and 17.3% (*n* = 9) as being of Black race. Most patients were hormone receptor-positive and HER2-negative and had earlier stage (I/II) breast cancer and advanced (II/III) grade tumors. For body composition types, 42.3% (*n* = 22) of patients were considered adequate, 26.9% (*n* = 13) had high adiposity, 25.0% (*n* = 14) had low muscle, and 5.8% (*n* = 3) had high adiposity with low muscle. Descriptive statistics of body composition components are given in Appendix A. The high adiposity group had the highest TAT as well as TSM (total skeletal muscle) areas compared with the other three groups (*p* < 0.05) (Appendix A). Patients having high adiposity and high adiposity with low muscle tended to have a higher proportion of very low-density muscle and low-density muscle compared with patients having low muscle and adequate body composition types, although these differences were not statistically significant (Appendix A). Patients with high adiposity were less likely to have stage III disease (7.7%, *n* = 1) compared with patients with adequate (18.2%, *n* = 4) or low muscle (21.4%, *n* = 3) type (Appendix A). Appendix A shows the comparison of body composition types with the clinical definition of obesity (BMI ≥ 30 kg/m^2^) and sarcopenia (skeletal muscle index [SMI], skeletal muscle area/height in meters squared, <38.9 cm^2^/m^2^) among patients without missing data on weight and height (*n* = 43; 82.7% of 52). The highest proportion of obesity was in patients with the high adiposity body composition type (81.1%, *n* = 9). Among five patients with sarcopenia, three (60%) were in the low muscle and one (20%) was the in high adiposity with low muscle groups.

For 168 patients with breast cancer and CT scan-assessed body composition, there were suggestive associations of low muscle (hazard ratio [HR] = 2.21, 95% confidence interval [CI] = 0.79–6.19) and low muscle with high adiposity (HR = 1.78, 95% CI = 0.37–8.46) types with survival, compared with the adequate type (Appendix A). These estimates were not statistically significant, potentially due to the small number of outcomes (25 deaths total).

Figure 1 shows the intra-patient and inter-tissue concordance of markers. For markers in the epithelial compartment, the intra-patient correlation (r) ranged between 0.88 and 0.58, suggesting a high to moderate level of similarity between the two regions of interest (ROIs) from a single patient. Markers in the stromal compartment remained high (r ranging from 0.82 to 0.35), although the concordance levels were generally lower than those in the epithelial compartment. Specifically, CTLA4 expression had a consistently high correlation within the tumor and stromal compartments and between the two compartments. ER-α and HER2 had a high correlation in the tumor compartment, whereas cluster of differentiation 8 (CD8) was among those with a high correlation in the stromal compartment. By contrast, forkhead box P3 (FOXP3) and pan-AKT had lower correlations among the markers.

A cluster analysis showed a clear distinction between markers in the epithelial vs. stromal components (Figure 2). The levels of proteins indicating tumor epithelium (e.g., pan cytokeratin [CK]) and stroma (e.g., fibronectin and α-smooth muscle actin [SMA]) differed significantly between these two components (Table 2). The levels of proteins in epithelium measured by the GeoMx Digital Spatial Profiler (DSP) were consistently higher in ER-positive, PR-positive, and HER2-positive tumors compared with ER-negative, PR-negative, and HER2-negative tumors defined by pathology reports, although only the HER2 result was statistically significant (*p* < 0.001) (Appendix A). 

Unsupervised clustering analysis did not show a clear profile for the markers by body composition type (Appendix A). However, low muscle (vs. adequate) was associated with lower pan-AKT (log2-fold change = −0.92; *p* = 0.0427; full model) expression in tumor (Figure 3; Table 3). Low muscle was also associated with higher CTLA4 expression in tumor (log2-fold change = 0.53, *p* = 0.0589). Low muscle was associated with higher expression of inositol polyphosphate-4-phosphatase type II B in stroma (INPP4B; log2-fold change = 1.18, *p* = 0.0003. FDR correct *p* = 0.0280).

High adiposity was associated with high expression of CD8 (log2-fold change = 1.16, *p* = 0.0429), CD45RO (log2-fold change = 0.95, *p* = 0.0425), and CD20 (log2-fold change = 0.61, *p* = 0.0314). High adiposity was also associated with higher expression of INPP4B, GZMB, and CD3 in the stroma, although the associations were not significant in the full models. The levels of these protein markers by clinical characteristics are given in Appendix A. 

The high adiposity with low muscle type was associated with lower expression of phospho-tuberin (T1462) (log2-fold change = −1.52, *p* = 0.0205) in tumor epithelium and phospho-PRAS40 (T246) in both tumor epithelium (log2-fold change = −1.96, *p* = 0.0338) and stroma (log2-fold change = −1.36, *p* = 0.0512). This body composition type was also associated with higher expression of CTLA4 (log2-fold change = 1.10, *p* = 0.0383), but lower expression of PD-1 (log2-fold change = −1.22, *p* = 0.048), CD14 (log2-fold change = −2.18, *p* = 0.0231), and phospholipase C gamma 1 (PLCG1; log2-fold change = −1.30, *p* = 0.0264) in stroma. These findings, as well as findings from the low muscle type and the high adiposity type, were unchanged after additionally adjusting for receipt of chemotherapy (Appendix A). 

Additional analyses using tertiles of TSM results were consistent with the body composition types we proposed. For example, the estimate of the association (log2-fold change) of pan-AKT in the tumor compartment was –0.78 (*p* = 0.057) for the first tertile vs. the second and third tertiles of the TSM, and –0.92 (*p* = 0.043) for low muscle vs. the adequate type (Appendix A). Similarly, tertiles of TAT results showed associations consistent with the high adiposity vs. adequate body composition groups (Appendix A). The associations for high adiposity vs. adequate groups were stronger compared with those for the tertiles of WC (Appendix A) and BMI groups (Appendix A). 

## 3. Discussion

In this cross-sectional study, we investigated the extent to which body composition type, classified by CT scan image-assessed adipose and muscle tissue areas, was associated with protein expression levels in the immune and PI3K/AKT signaling pathways in breast cancer. We used a high-plex, spatially resolved method to assess tumor epithelium and CD45-rich stromal microenvironment. Several associations were identified for the low muscle and high adiposity body composition types, and these associations were stronger than those of WC or BMI. The associations observed in this study should be considered hypothesis-generating because of the small sample size. 

Among body composition components, low muscle is considered an important factor in addition to high adiposity in increasing mortality risk among patients with cancer [17,18], and an explanation is the influence of low skeletal muscle mass on the tumor immune microenvironment [9]. Data on the extent to which body composition components are related to tumor immune modulation in breast cancer are very limited. Research findings on BMI and TIL levels have proven inconsistent, as studies reported a positive association [19], a negative association in patients who received neoadjuvant chemotherapy [5], and no association in triple-negative breast cancer [20]. Only one study investigating specific immune cells reported that BMI changes since age 18 were associated with increased CD4 and CD163 levels in breast cancer [21]. Data from other high immune activity cancer types, such as kidney cancer, may shed light on this area. Ged et al. reported that in 62 patients with metastatic clear cell renal cell carcinoma in TCGA, low SMI was associated with a higher degree of macrophage infiltration and T helper type 1 cells in tumors, estimated using RNA immune deconvolution methods [7]. However, in the same study, SMI was not associated with total immune infiltration or gene expression of immune checkpoint markers, including CTLA4, programmed death ligand 1 (PD-L1), and PD-1. The algorithm-based method has a limitation of precisely measuring these immune checkpoint markers. In our study, low muscle mass, represented by the high adiposity with low muscle and low muscle body composition types, was associated with higher expression of CTLA4 protein. In a study of patients with non-small cell lung cancer receiving PD-1 checkpoint blockade therapy, higher (vs. lower) expression of CTLA4 protein in the macrophage compartment assessed by the DSP assay was associated with worse survival [22]. Thus, these data suggest that immune inhibition in tumors can be an intermediate factor of low muscle mass adversely affecting patient prognosis. Our samples were mainly ER+ tumors, a subtype with lower levels of TIL compared to ER- or triple-negative tumors [23]. However, ER+ tumors form a heterogeneous group, and the immune landscape, including stromal TILs, CD8+ T-cells, and PD-L1, in ER-low and intermediate tumors (ER expression ≤ 50% of the cells) are similar to that of primary triple-negative tumors [24]. Higher TILs and CD8+ T-cell exhaustion are associated with worse survival among patients with ER+ tumors [23,25], suggesting that immune checkpoints may be an important mechanism for this breast cancer subtype. A future research direction is to examine how the immune pathways interact with different types of endocrine therapy among patients with ER+ breast cancer [23].

Our finding on the PI3K/AKT pathway is consistent with our knowledge of cancer-related muscle mass reduction. We observed that tumor pan-AKT expression was lower in patients with the low muscle composition type. This finding was consistent with the lower tumor expression levels of phospho-tuberin T1462 and phospho-PRAS40 T246—both of which are substrates of AKT—in patients having the high adiposity with low muscle body composition type [26,27,28]. Reduced phospho-tuberin T1462 and phospho-PRAS40 T246 lead to higher activity of tuberin, i.e., tuberous sclerosis complex 2 (TSC2), and PRAS40, both of which are inhibitors of MTOR complex 1. In patients with breast cancer in TCGA, lower (vs. higher) expression of phospho-tuberin T1462 and phospho-PRAS40 T246 in luminal breast cancer is associated with better breast cancer-specific survival (Supplemental Appendix A). In addition, both the low muscle and high adiposity types were associated with higher expression of INPP4B in the stromal compartment. INPP4B, similar to phosphatase and tensin homolog (PTEN), dephosphorylates and in turn, obstructs AKT downstream signaling [29,30,31]. The association of lower activity of PI3K/AKT and potentially MTOR with the low muscle body composition type is in line with the observation that patients with cancer are at high risk of losing muscle, leading to sarcopenia or cachexia [32]. Our finding for the tumor would need to be verified by testing PI3K/AKT and MTOR activities in skeletal muscle tissue. A study has shown mammary tumor-induced skeletal muscle dysfunction via interleukin-15 gene suppression in mice and a similar result for patients with breast cancer [33].

A limitation of the present study is that the CT scans obtained spanned the time before and after breast cancer diagnosis and surgery. Approximately half of the patients only had CT scans available at a time after breast cancer surgery, which was the source of our tissue sample. This situation may have resulted in the bias of reversed causality. Also, among some patients, the time between their CT scan available for this study and breast cancer surgery was very long, as we were unable to restrict the time to a specific range. We assumed that the CT scan-assessed body composition was a proxy for the true body composition at the time of breast cancer diagnosis. However, body composition might have changed during treatments for breast cancer [34,35]. We previously showed that the body composition assessed by opportunistic CT scans during a similar time period had high intra-individual reliability in general, although the TAT area on L3 images decreased by 10% while the decrease was smaller for skeletal muscle area (−1.27%) [36]. For patients with body composition assessed on a CT scan taken in the later stage of the treatment period, their adipose and muscle tissue areas may have been underestimated, resulting in a measurement error that may have affected the associations toward null. 

While we accounted for important demographic and clinical characteristics using regression methods, we were unable to account for other unmeasured potential confounders, including local adipose tissue, neoadjuvant chemotherapy, and menopausal status. As the early stage of invasive cancer develops from the epithelium of ducts or lobules of the breast, it affects the surrounding adipocytes. These cancer-associated adipocytes are remodeled and reversely modify the cancer cells, leading to a more aggressive behavior [37,38]. However, the relative contribution of local adipocytes vs. distal adipose tissue (e.g., visceral adipose tissue) to the breast microenvironment is unclear. Neoadjuvant chemotherapy may affect immune characteristics in tumors obtained from surgery for breast cancer treatment. We were unable to account for the potential influence of neoadjuvant chemotherapy, although the overall chemotherapy status, which included any chemotherapy treatment, did not change our findings. In addition, the influence of adipose tissue on tumors may vary by menopausal status. Obesity (BMI ≥ 30 kg/m^2^) is associated with increased breast cancer risk in postmenopausal women [39], which is likely associated with the excessive estrogen secreted by adipose tissue. However, central obesity (high WC) is associated with increased breast cancer risk in both pre- and post-menopausal women, suggesting that other cancer promotors, including adipokines and insulin-like growth factors, from central adipose tissue may not differ by menopausal status [39], Future research should consider these factors to improve study validity. 

The present study has several additional limitations. We had limited ability to account for potential heterogeneity in tumor tissue because the DSP assay was performed in a tissue microarray core from each patient. Performing a DSP assay with multiple areas of interest on a slide containing a whole section of the tumor would improve the representativeness of assayed areas. The small sample size in this study, designated for an exploratory study, limited the statistical power. Our post hoc power analysis showed that almost all markers were underpowered (<80%), except for INPP4B in the stromal compartment (Appendix A). Markers with low statistical power and null association in this study should not be excluded in future investigations. Our findings have limited generalizability because they were based on opportunistic CT scans, which are prescribed more often for patients with stage II or higher cancer than stage 0 or I cancer. In addition, patients with eligible tumor samples on a tissue microarray had more recently received a diagnosis than patients without a tumor sample or with ineligible samples (Appendix A). 

In conclusion, this study showed several promising associations of adipose and skeletal muscle tissue body composition components with the PI3K/AKT and immune pathways in patients with breast cancer. Specifically, high adiposity is associated with higher expression of several types of immune cells, and there is a potential link between low muscle mass and increased immune checkpoint (CTLA4) as well as AKT inhibition. If confirmed, these findings will improve our understanding of tumor biology related to body composition and indicate the potential utilization of the information for predicting treatment efficacy and patient outcomes. 

## 4. Materials and Methods

Patient population

Women who were diagnosed with invasive breast cancer and received breast cancer treatment at the University of Florida Health Shands Hospital from October 2011 to April 2020 were identified through the local tumor registry linked to the electronic medical record system. Appendix A shows the flowchart of patient selection. Eligible participants were women 20–75 years of age at diagnosis with an archived abdominal or pelvic CT scan. For patients with multiple CT images, scans prior to and closest to breast cancer surgery were selected. We excluded patients with a history of diabetes, as this disorder and its treatment can affect the activity of the PI3K-AKT pathway [40,41], and patients who were pregnant at the time of the CT scan. Of 529 eligible patients, 152 (28.7%) had surgical formalin-fixed, paraffin-embedded tumor blocks available for review, and 57 (10.8%) had tumor blocks eligible for tissue microarray construction (tumor diameter > 2 mm). Demographic and clinicopathological variables were extracted from the tumor registry and electronic medical records. Height and weight measured nearest the time of CT scan were extracted for BMI calculation. Data from 52 patients with complete tumor protein assay and CT scan-assessed body composition were used in the final statistical analyses. Appendix A shows the time interval (days) between CT scan procedure and breast cancer diagnosis and between CT scan and surgery, i.e., the time when the tumor samples were obtained. Approximately half of the participants had an eligible CT scan prior to surgery. We have shown that adiposity and muscle areas annotated on CT scans performed in a similar period of time have high intra-individual reliability (intraclass correlation coefficients ranged from 0.763 to 0.998) in a subsample of patients with breast cancer derived from the same population used in the current study [36].

Body composition measurements 

The method used for annotating muscle and adipose tissue on clinical CT scans has been described in detail elsewhere [42]. Briefly, CT images at the third lumbar vertebra (L3) were analyzed using SliceOmatic, version 5.0 revision 7 (Tomovision, Montreal, QC, Canada). We used pre-defined Hounsfield unit thresholds and the anatomical location (Appendix A) [43,44] of adipose and muscle to annotate three adipose tissue components—visceral adipose tissue, subcutaneous adipose tissue, and intermuscular adipose tissue—and five muscle density levels: very low-density muscle, low-density muscle, normal-density muscle, high-density muscle, and very high-density muscle. To classify body composition types, the area of TAT was calculated as the summation of subcutaneous adipose tissue, intermuscular adipose tissue, and visceral adipose tissue areas; the TSM area was calculated as the summation of the areas for all five levels of muscle density. We categorized patients into four mutually exclusive body composition types based on tertiles of TAT and TSM: (1) adequate, defined as patients with the highest two tertiles of TSM and the lowest two tertiles of TAT; (2) high adiposity, defined as the highest tertile of TAT and the highest two tertiles of TSM; (3) low muscle, defined as the lowest tertile of TSM and the lowest two tertiles of TAT; and (4) high adiposity with low muscle, defined as the third tertile of TAT and the first tertile of TSM. The cutoffs for tertiles were 368.34 cm^2^ and 522.15 cm^2^ for TAT, and 116.54 cm^2^ and 131.65 cm^2^ for TSM. We used this classification because, among female patients with colorectal cancer, those who have the high adiposity with low muscle type have a 64% higher risk of overall mortality than those who have the adequate type [45]. Similarly, a study assessing patients with nonmetastatic breast cancer showed that compared with the first tertile of TAT plus non-sarcopenia, the third tertile of TAT plus sarcopenia is associated with a two-fold increase in risk of death [1]. The body circumference measured on L3 images was used as a proxy for WC. 

GeoMx Digital Spatial Profiler (DSP) analysis

Tissue microarray construction was guided by board-certified breast pathologists (S.M.F. and J.R.A.). One tumor core (1.5 mm in diameter) was selected per patient. A 5 µm section of tissue microarray was freshly cut, processed for antigen retrieval, and incubated with primary antibodies covalently bonded with indexing oligonucleotide barcodes [46]. For the antibodies, we selected four protein modules: Immune Cell Profiling Core, Immune Cell Typing Panel, PI3K/AKT Signaling Panel, and Pan-tumor Panel (complete list of 43 antibodies in Figure 1) for this experiment. After the incubation, slides were scanned with a GeoMx DSP instrument (NanoString Technologies, Seattle, WA, USA), and two ROIs, each 300 µm in diameter, were drawn on a tumor core enriched for pan-cytokeratin (panCK) and CD45, the morphological markers (Figure 4). Automated segmentation was performed to identify regions expressing panCK as tumor epithelial cells and the rest of the region as the CD45-rich stromal microenvironment. The indexing oligonucleotides in the ROIs were then cleaved from the antibodies using UV light, collected via microcapillary aspiration, and digitally counted using the nCounter system. Digital counts from barcodes corresponding to protein were normalized by geometric means of housekeeping proteins (S6, histone H3, and GAPDH) and corrected for non-specific binding of antibodies on tissue estimated by the mean counts of mouse IgG1, mouse IgG2a, and rabbit IgG. The background-corrected values were then transformed using log2 values for statistical analyses. All proteins passed the internal quality control process set by NanoString.

Statistical analysis

The normality of continuous variables was examined using Shapiro–Wilk tests. Patient demographic and clinical characteristics were summarized overall and across the body composition types. Descriptive analyses were also performed for specific adipose and muscle tissue areas. Intra-patient, i.e., ROI 1 vs. ROI 2, and inter-tissue, i.e., epithelium and stroma, concordance of markers was assessed using Pearson’s correlation coefficients. Principal component analysis clustering according to tissue compartments and body composition type was performed using the *prcomp* function in R (version 4.2.2) and plotted using the package ggbiplot (version 0.6.2). Heatmaps were generated using the *heatmap.2* function in the R package gplots (version 3.2.0). The association of body composition types with protein markers was assessed by linear mixed models, separately for epithelial and stromal components. Two models were performed, a basic model adjusting for analytical batch, and a full model adjusting for analytical batch, race, breast cancer stage, and tumor grade. In the models, an individual was assigned as the random effect, with the assumption that data from the same individual were correlated. For immune and PI3K/AKT pathway proteins, *p*-values were corrected for multiple comparisons using the Benjamini–Hochberg false discovery rate (FDR; at 0.2) method. Volcano plots were generated using the *ggplot* function in the R package ggplot2 (version 3.5.1). Several sensitively analyses were performed for significant associations. The regression models were additionally adjusted for chemotherapy receipt (ever or never). In addition, analyses were performed to compare protein expression levels between high adiposity vs. adequate adiposity, represented by the third tertile vs. the second and first tertiles of TAT, and low muscle mass vs. adequate muscle mass, represented by the first tertile vs. the second and third tertiles of TSM. Protein expression levels of each tertile group for both TAT and TSM were also compared. Those models adjusted TAT and TSM mutually, in addition to other covariates. To compare the utility of the body composition types with anthropometric measurement of body fatness in predicting tumor protein expression, we modeled comparing high (tertile 3) vs. low (tertile 1) areas of WC measured on CT scans and comparing overweight and obesity (vs. normal weight) defined by BMI among patients with data available. Additional queries on the survival analysis of protein markers were performed using The Cancer Proteome Atlas breast cancer (TCPA-BRCA) [47] data using an online tool TRGAted (https://nborcherding.shinyapps.io/TRGAted/ accessed on 9 December 2024) [48]. The association of body composition type and overall survival was examined using the Cox proportional hazards models in SAS (SAS Institute, Cary, NC, USA; version 9.4) for 168 patients with annotated CT scans. All tests were two-sided. Post hoc power analysis was performed using *simr* (version 1.0.7) in R. The program estimated statistical power before and after including the body composition type variable in the fully adjusted linear mixed models. 

## Figures and Tables

**Figure 1 ijms-25-13428-f001:**
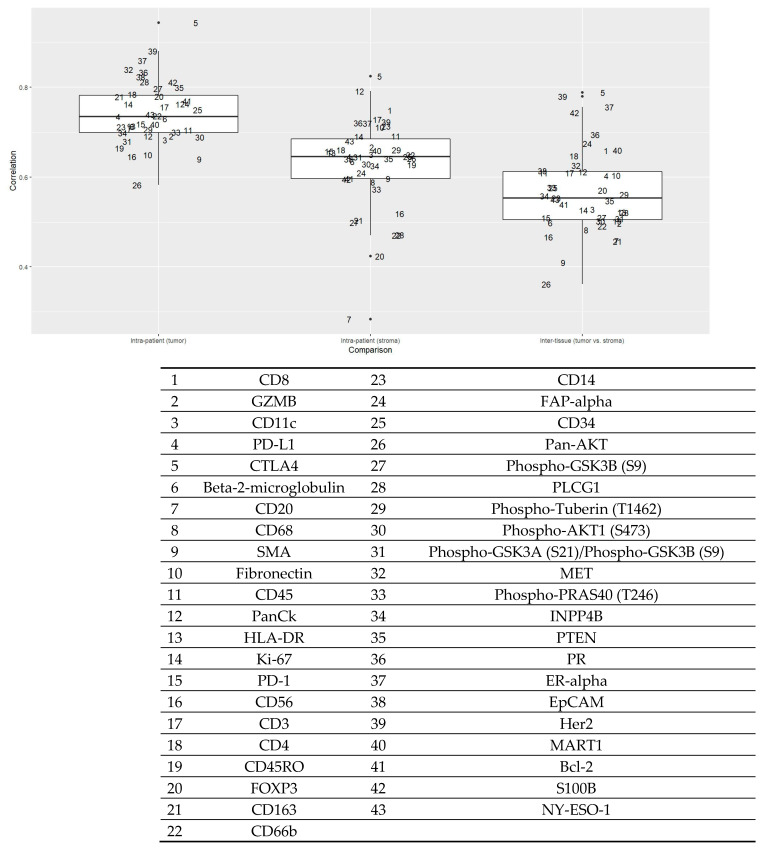
Intra-patient (epithelial and stromal components) and inter-tissue concordance of proteins (*n* = 52 patients). The Y-axis is Pearson’s correlation coefficient. The box plot on the left shows the correlation of each marker in the tumor compartment within patients. The box plot in the middle shows the correlation of each marker in the stroma compartment within patients. The box plot on the right shows the correlation of each marker between the tumor and stromal components. Each number represents a protein given in the table below.

**Figure 2 ijms-25-13428-f002:**
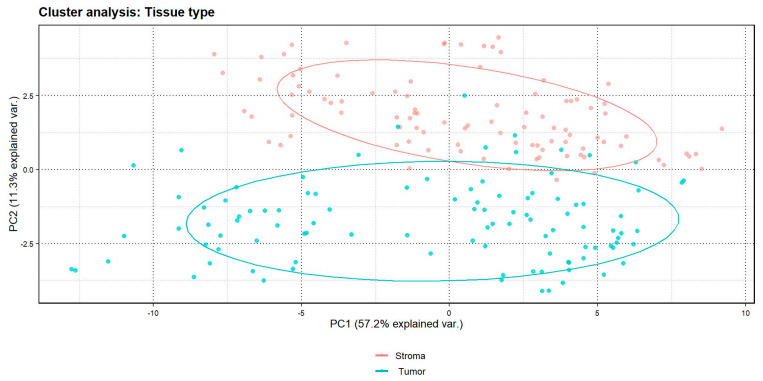
Cluster analysis by tissue type (tumor epithelium vs. stroma); *n* = 52 patients. The “% explained var”. in the X-axis and Y-axis represents the percentage of variance explained.

**Figure 3 ijms-25-13428-f003:**
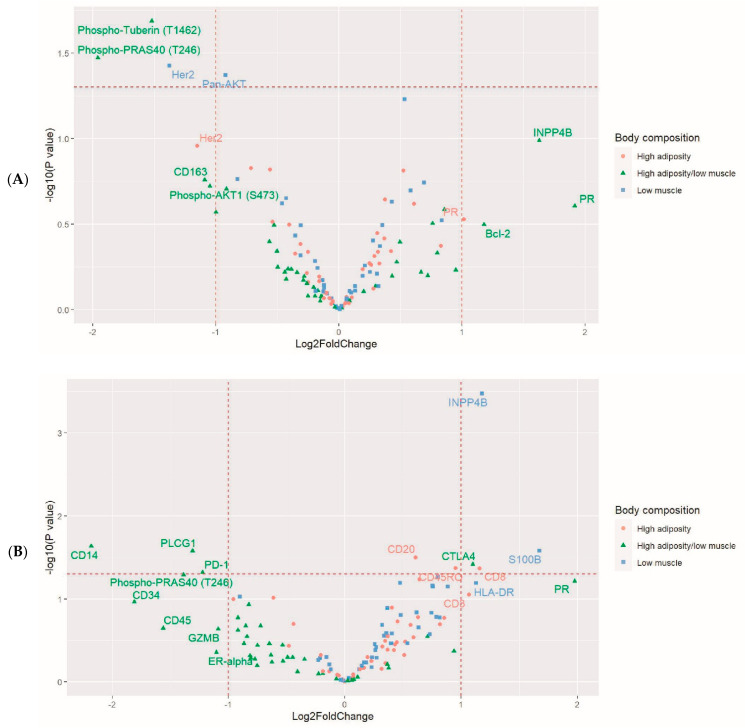
Volcano plots for the associations of body composition type with proteins in tumor (**A**) epithelium and (**B**) stroma. The horizontal dot lines indicate *p*<0.05. The vertical dot lines indicate a two-fold increase or decrease. Multivariable models adjusted for analytical batch, race, breast cancer stage, and tumor grade; *n* = 52 patients.

**Figure 4 ijms-25-13428-f004:**
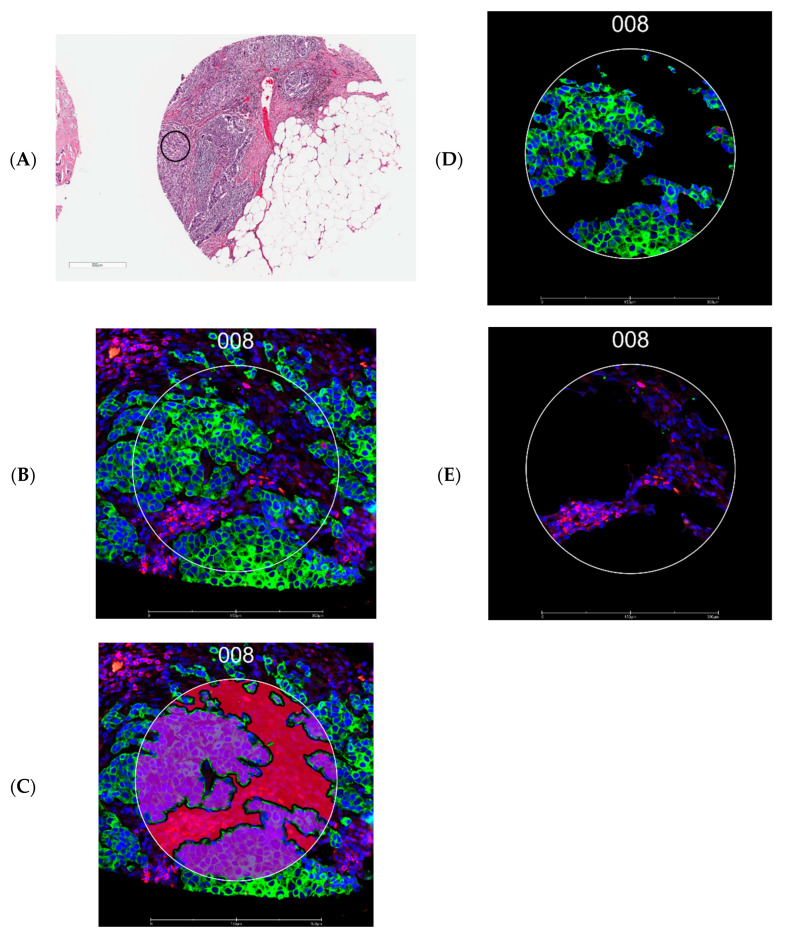
Representative images from tissue microarrays. (**A**) Region of interest in a tissue microarray core (black circle indicates the approximate location). (**B**) Fluorescence image of the region of interest selected for tumor (guided by panCK, green) and stromal areas enriched for leukocytes (guided by CD45, red) identified using the GeoMx Digital Spatial Profiler. (**C**) Segmentation by morphology marker (panCK). (**D**) The epithelial segment. (**E**) The stromal segment; 008 represents an ROI number. Scale bar: 500 µm for A and 300 µm for (**B**–**D**).

**Table 1 ijms-25-13428-t001:** Patient demographic and clinical characteristics (total *n* = 52).

Characteristic	Mean (SD) or No. %
Age at diagnosis (years)	56.8 (10.5)
Age at CT scan (years)	57.2 (10.3)
Waist circumference (cm)	99.4 (13.9)
**Race**	
White	43 (82.7)
Black	9 (17.3)
**Cancer stage**	
I	17 (32.7)
II	26 (50.0)
III	9 (17.3)
**Tumor grade**	
I	5 (9.7)
II	28 (53.8)
III	19 (36.5)
**Tumor size**	
<25 mm	24 (46.2)
≥25 mm	17 (32.7)
Missing	11 (21.1)
**ER status**	
Positive	46 (88.5)
Negative	6 (11.5)
**PR status**	
Positive	43 (82.7)
Negative	9 (17.3)
**HER2 status**	
Positive	10 (19.3)
Negative	41 (78.8)
Missing	1 (1.9)
**Molecular subtype** ^a^	
HR+/HER2+	9 (17.3)
HR+/HER2-	37 (71.2)
HR-/HER2+	1 (1.9)
HR-/HER2-(triple-negative)	4 (7.7)
Missing	1 (1.9)
**Chemotherapy**	
Yes	24 (46.2)
No	27 (51.9)
Missing	1 (1.9)
**Body composition**	
Adequate	22 (42.3)
High adiposity	13 (25.0)
Low muscle	14 (26.9)
High adiposity with low muscle	3 (5.8)

^a^ HR, hormone receptor, i.e., ER and PR. HR+ includes ER+, PR+, or both tumors. HR-includes ER- and PR-tumors. Abbreviations: CT, computed tomography; ER, estrogen receptor; PR, progesterone receptor.

**Table 2 ijms-25-13428-t002:** Distribution of protein markers in tumor and stromal compartments, sorted by FDR-corrected *p*-values.

Marker	Tumor ^a^, Median (IQR)	Stroma ^a^, Median (IQR)	*p*-Value	FDR-Corrected *p*-Value
PanCk	9.13 (2.6)	5.33 (2.06)	3.47 × 10^−38^	1.49 × 10^−36^
ER-alpha	5.76 (3.58)	2.66 (3.14)	1.88 × 10^−28^	4.04 × 10^−27^
Phospho-GSK3B (S9)	2.41 (3.32)	0 (1.21)	5.31 × 10^−22^	7.61 × 10^−21^
EpCAM	3.71 (3.13)	1.28 (2.1)	1.77 × 10^−21^	1.90 × 10^−20^
Fibronectin	6.89 (3.31)	9.39 (3.04)	1.34 × 10^−20^	1.15 × 10^−19^
Her2	3.48 (2.33)	1.87 (2.12)	4.60 × 10^−19^	3.30 × 10^−18^
INPP4B	5.06 (2.87)	3.16 (2)	9.51 × 10^−19^	5.84 × 10^−18^
PR	3.69 (4.17)	1.63 (2.05)	1.51 × 10^−18^	8.12 × 10^−18^
MET	2.59 (3.05)	1.31 (2.2)	2.64 × 10^−16^	1.26 × 10^−15^
Bcl-2	4.58 (2.51)	3.37 (1.87)	3.30 × 10^−13^	1.42 × 10^−12^
SMA	8.5 (2.38)	10.16 (2.03)	5.92 × 10^−11^	2.31 × 10^−10^
NY-ESO-1	1.83 (3.24)	0.79 (2.32)	1.92 × 10^−10^	6.88 × 10^−10^
Phospho-GSK3A (S21)/Phospho-GSK3B (S9)	2.76 (2.8)	1.26 (1.67)	3.06 × 10^−10^	1.01 × 10^−9^
Ki-67	2.7 (2.62)	2.25 (1.98)	4.18 × 10^−8^	1.28 × 10^−7^
FOXP3	0.26 (2.24)	0 (0.85)	1.06 × 10^−7^	3.04 × 10^−7^
Pan-AKT	5.92 (2.42)	4.95 (1.78)	1.80 × 10^−7^	4.84 × 10^−7^
CD45	3.53 (2.7)	4.46 (2.36)	2.08 × 10^−7^	5.26 × 10^−7^
PLCG1	1.52 (2.71)	1.06 (2.06)	4.92 × 10^−7^	1.18 × 10^−6^
CD163	−0.02 (2.31)	0.23 (1.72)	1.12 × 10^−6^	2.41 × 10^−6^
MART1	0.9 (3.34)	0.1 (2.09)	1.11 × 10^−6^	2.41 × 10^−6^
HLA-DR	3.41 (3.13)	4.58 (2.79)	2.46 × 10^−6^	5.04 × 10^−6^
Phospho-PRAS40 (T246)	1.71 (3.18)	0.7 (2.23)	8.68 × 10^−6^	1.70 × 10^−5^
CD34	2.62 (2.99)	3.67 (2.74)	9.66 × 10^−6^	1.81 × 10^−5^
CD11c	2.91 (2.96)	3.69 (2.16)	6.63 × 10^−5^	0.000118788
CD3	1.32 (3.61)	2.26 (2.97)	7.21 × 10^−5^	0.000124012
FAP-alpha	0.16 (2.74)	1.06 (2.45)	0.000546	0.000903
Phospho-AKT1 (S473)	2.1 (2.96)	1.46 (2.02)	0.00173	0.002755185
CD66b	−0.08 (1.84)	0 (0.89)	0.00405	0.006219643
GZMB	3.39 (2.81)	3.59 (2.39)	0.0102	0.015124138
PD-1	1 (2.81)	0.95 (2.1)	0.0125	0.017916667
CD45RO	1.71 (2.38)	1.27 (2.41)	0.0139	0.019280645
CD20	0.85 (2.47)	0.79 (1.84)	0.0172	0.0231125
CD4	2.09 (2.95)	2.63 (2.83)	0.0252	0.032836364
Beta-2-microglobulin	2.52 (2.76)	2.3 (2.3)	0.0322	0.040723529
CTLA4	−0.02 (4.8)	0 (2.9)	0.0353	0.043368571
PD-L1	−0.11 (1.69)	−0.03 (1.05)	0.0418	0.049927778
CD14	2.63 (3.38)	2.95 (2.24)	0.0496	0.057643243
CD56	2.01 (2.95)	2.03 (2.14)	0.0638	0.07009
Phospho-Tuberin (T1462)	0 (2.15)	−0.04 (0.95)	0.0637	0.07009
PTEN	1.31 (3.36)	1.21 (2.53)	0.0652	0.07009
CD68	3.27 (2.82)	3.6 (1.64)	0.445	0.466707317
CD8	3.08 (3.15)	3.38 (2.81)	0.599	0.613261905
S100B	2.5 (3.65)	2.73 (4.1)	0.701	0.701

^a^ Protein expression is given as log2 values. Abbreviations: FDR, false discovery ratio; IQR, interquartile range.

**Table 3 ijms-25-13428-t003:** Associations between body composition type and immune and PI3K/AKT protein expression in breast tumors. Results with nominal *p* < 0.05 are included. *n* = 52 patients.

			Basic Model ^a^			Full Model ^b^		
Body Composition Type (Compared with Adequate Type)	Marker	Tissue Compartment	Log2-Fold Changes (95% CI)	*p* Value	FDR-Corrected *p* Value	Log2-Fold Changes (95% CI)	*p* Value	FDR Corrected *p*
Low muscle	Pan-AKT	Tumor	−0.76 (−1.63, 0.12)	0.1077	0.9756	−0.92 (−1.71, −0.13)	0.0427	0.8243
Low muscle	CTLA4	Tumor	0.60 (0.07, 1.28)	0.0362	0.9756	0.53 (0.04, 1.03)	0.0589	0.8244
Low muscle	INPP4B	Stroma	1.14 (0.59, 1.69)	0.0003	0.0270	1.18 (0.65, 1.72)	0.0003	0.0280
High adiposity	INPP4B	Stroma	0.87 (0.31, 1.43)	0.0054	0.2200	0.64 (0.06, 1.23)	0.0578	0.4168
High adiposity	CD8	Stroma	1.35 (0.38, 2.32)	0.0115	0.2200	1.16 (0.17, 2.15)	0.0429	0.4168
High adiposity	CD45RO	Stroma	1.10 (0.29, 1.91)	0.0131	0.2200	0.95 (0.14, 1.76)	0.0425	0.4168
High adiposity	GZMB	Stroma	1.02 (0.24, 1.80)	0.0170	0.2382	0.63 (−0.17, 1.43)	0.1656	0.6223
High adiposity	CD20	Stroma	0.52 (0.03, 1.02)	0.0434	0.3397	0.61 (0.09, 1.13)	0.0314	0.4168
High adiposity	CD3	Stroma	1.14 (0.10, 2.18)	0.0440	0.3397	1.07 (−0.02, 2.16)	0.0884	0.4950
High adiposity/low muscle	Phospho-tuberin (T1462)	Tumor	−1.08 (−2.28, 0.12)	0.0946	0.9900	−1.52 (−2.67, −0.39)	0.0205	0.4735
High adiposity/low muscle	Phospho-PRAS40 (T246)	Tumor	−1.41 (−3.11, 0.28)	0.1197	0.9900	−1.96 (−3.59, −0.36)	0.0338	0.4735
High adiposity/low muscle	Phospho-PRAS40 (T246)	Stroma	−1.36 (−2.60, −0.11)	0.0445	0.3397	−1.39 (−2.62, −0.16)	0.0512	0.4168
High adiposity/low muscle	CTLA4	Stroma	0.98 (0.09, 1.88)	0.0436	0.3397	1.10 (0.19, 2.02)	0.0383	0.4168
High adiposity/low muscle	PD-1	Stroma	−0.98 (−2.05, 0.08)	0.0873	0.4398	−1.22 (−2.29, −0.16)	0.0480	0.4168
High adiposity/low muscle	CD14	Stroma	−2.43 (−4.13, −0.73)	0.0096	0.2200	−2.18 (−3.82, −0.54)	0.0231	0.4168
High adiposity/low muscle	PLCG1	Stroma	−1.28 (−2.37, −0.19)	0.0315	0.3397	−1.31 (−2.36, −0.26)	0.0264	0.4168

^a^ Basic model adjusted for analytical batch. ^b^ Full model adjusted for analytical batch, race, breast cancer stage, and tumor grade.

## Data Availability

Data will not be available to the public to protect patient privacy. The R codes can be accessed at DOI 10.5281/zenodo.10647149. The record is publicly accessible, but files are restricted to users with access.

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
