# Peer review of "Association of Computed Tomography Scan-Assessed Body Composition with Immune and PI3K/AKT Pathway Proteins in Distinct Breast Cancer Tumor Components"

_ijms, 2024, doi:10.3390/ijms252413428_

Round 1
Reviewer 1 Report
Comments and Suggestions for Authors
Comments and Suggestions for Authors
In the submitted manuscript authors assessed the extent to which computed tomography-assessed body composition phenotypes are associated with immune and PI3K/AKT signaling pathways in breast tumors, and showed that, compared with the normal type, the low muscle type was associated with higher expression of INPP4B, while other significant associations included low muscle type with increased CTLA4 and decreased pan-AKT expression in tumor epithelium, and high adiposity with increased CD3, CD8, CD20, and CD45RO expression in stroma.
This manuscript is quite well written and study behind it seems robust; however, there are several important things which must be corrected or further improved:
1) Supplementary material must be consecutively numbered according to its appearance in the text, so Supplemental Table S4 cannot be mentioned before ST1-3, and Supplemental Figure S2 cannot be mentioned before SFS1, etc.
2) For each figure it must be clear what it presents (through explanation in figure legend) and how it was presented (both figure legend and proper names of Y- and X-axes), since for example, it is unknown how “concordance” was presented in Figure 1, since it is showing unspecified “correlation”, while it is also unclear how data were presented! In figure legend also sample size must be provided. And, by the way, figure legends are written BELOW figures.
3) It is unclear why continuous variables were presented with mean and SD, without any proof that they follow normal distribution, and then also minimum and maximum values were presented for some (and then where is the median?!).
4) Whenever N is presented in table or text, also percentage must be stated, and vice versa.
5) Not a single statistical test for data comparison has been mentioned in “Statistical analysis” section, while it is again unclear why ad hoc parametric Pearson’s correlation coefficient and (out of nowhere) t-test (SFS2) were used?! This applies to all statistical analyses/tests.
6) For ALL R-packages, the used version number must be stated and proper reference cited (if published in scientific journal).
7) For ALL on-line tools and databases, valid URL must be provided and proper reference cited, while for TCGA data, it must be precisely stated which dataset was actually used (I assume TCGA-BRCA) and its sample size, while also The Cancer Proteome Atlas must be cited.
8) Creating company of SAS software must be stated.
9) It is unclear why (results of) statistical analysis of Supplementary Table S5, S6, S7 were not conducted/presented?!
10) Authors must re-check text that all abbreviations were explained after their FIRST mentioning (e.g., ROI).
Author Response
1. Supplementary material must be consecutively numbered according to its appearance in the text, so Supplemental Table S4 cannot be mentioned before ST1-3, and Supplemental Figure S2 cannot be mentioned before SFS1, etc.
Response: Thank you very much for the opportunity to respond to the comments and revise the manuscript. We have now highlighted the changes in the manuscript. We have now made sure the Supplementary material is consecutively numbered according to its appearance in the text.
2. For each figure it must be clear what it presents (through explanation in figure legend) and how it was presented (both figure legend and proper names of Y- and X-axes), since for example, it is unknown how “concordance” was presented in Figure 1, since it is showing unspecified “correlation”, while it is also unclear how data were presented! In figure legend also sample size must be provided. And, by the way, figure legends are written BELOW figures.
Response: We have now revised the figures and legends to present the figure clearly and added sample size. For example, the Figure 1 legend now reads “Figure 1. Intra-patient (epithelial and stromal components) and inter-tissue concordance of proteins (N=52 patients). The Y-axis is Pearson’s correlation coefficient. The box plot on the left shows the correlation of each marker in the tumor compartment within patients. The box plot in the middle shows the correlation of each marker in the stroma compartment within patients. The box plot on the right shows the correlation of each marker between the tumor and stromal components. Each number represents a protein given in the table below.”
We have now placed figure legends below figures.
3. It is unclear why continuous variables were presented with mean and SD, without any proof that they follow normal distribution, and then also minimum and maximum values were presented for some (and then where is the median?!).
Response: We tested normality using the Shapiro-Wilk test. The variables presented in Table 1, including age at diagnosis (P=0.13), age at CT scan (P=0.14), and waist circumference (P= 0.56) were normally distributed. However, body composition components (Supplemental Table S1) and time between CT scan to surgery variables (Supplemental Table S12) were somewhat skewed (P<0.05). Thus, we present both mean with SD and median with interquartile rage for these variables. We removed the minimum and maximum to avoid confusion. We have now added Shapiro-Wilk test in the Method section. (page 24)
4. Whenever N is presented in table or text, also percentage must be stated, and vice versa.
Response: We have now presented both N and percentage in tables (Table 1, Supplemental Table S2, and Supplemental Table S3) and corresponding text.
5. Not a single statistical test for data comparison has been mentioned in “Statistical analysis” section, while it is again unclear why ad hoc parametric Pearson’s correlation coefficient and (out of nowhere) t-test (SFS2) were used?! This applies to all statistical analyses/tests.
Response: We stated all statistical tests. We used Pearson’s correlation coefficient to show the concordance of markers within and between patients (p.24). We did not use a t-test. Regression analysis was performed using linear mixed models and Cox proportional hazard models.
6. For ALL R-packages, the used version number must be stated and proper reference cited (if published in scientific journal).
Response: We have now added version numbers to all R packages.
7. For ALL on-line tools and databases, valid URL must be provided and proper reference cited, while for TCGA data, it must be precisely stated which dataset was actually used (I assume TCGA-BRCA) and its sample size, while also The Cancer Proteome Atlas must be cited.
Response: We have added a valid URL for online tools. We have specified the TCPA dataset (BRCA) and cited a reference.
8. Creating company of SAS software must be stated.
Response: We have now stated the creating company of SAS (SAS Institute, Cary, NC). (p. 25)
9. It is unclear why (results of) statistical analysis of Supplementary Table S5, S6, S7 were not conducted/presented?!
Response: We did not perform statistical tests for Supplementary Tables S5 and S6 (now S2 and S3) because the sample size in each cell was small. Supplementary Table S7 (now S4) were results from Cox proportional hazard models.
10. Authors must re-check text that all abbreviations were explained after their FIRST mentioning (e.g., ROI).
Response: We have checked that all abbreviations were explained when they were first mentioned.
Reviewer 2 Report
Comments and Suggestions for Authors
Dear Authors,
I am pleased to provide my review of the manuscript titled *"Association of Computed Tomography Scan-Assessed Body Composition with Immune and PI3K/AKT Pathway Proteins in Distinct Breast Cancer Tumor Components."* This study addresses a topic of significant clinical importance, employing computed tomography (CT) imaging to investigate the relationship between body composition and key signaling pathways in breast cancer. The manuscript provides valuable insights that contribute to our understanding of tumor biology and patient outcomes.
However, there are areas where the manuscript could be strengthened to enhance its impact and clarity:
1. The time intervals between CT scans and tumor sampling may introduce variability in body composition assessments. A more detailed discussion of how this variability could bias the results is warranted.
2. Expand the discussion to highlight how the findings align with or diverge from existing literature, particularly concerning the roles of sarcopenia, adiposity, and immune modulation in breast cancer.
3. The inclusion of a power analysis is commendable. However, incorporating a concise summary of underpowered markers in the main text would help readers better contextualize the study’s findings.
4. Address minor typographical errors and ensure consistency in terminology throughout the manuscript (e.g., standardizing the use of "CTLA4" versus "CTLA-4").
Addressing these recommendations will improve the manuscript's clarity and strengthen its contribution to the field.
Best Regards
Author Response
- The time intervals between CT scans and tumor sampling may introduce variability in body composition assessments. A more detailed discussion of how this variability could bias the results is warranted.
Response: Thank you very much for the opportunity to respond to the comments. We have now highlighted the changes in the manuscript. We have discussed the variability of time intervals between CT scans and tumor sampling in detail as follows:
“A limitation of the present study is that the CT scans obtained spanned the time before and after breast cancer diagnosis and surgery. Approximately half of the patients only had CT scans available at a time after breast cancer surgery, which was the source of our tissue sample. The situation may have resulted in the bias of reversed causality. Also, among some patients, the time between their CT scan available for this study and breast cancer surgery was very long, as we were unable to restrict the time to a specific range. We assumed that the CT scan-assessed body composition was a proxy for the true body composition at the time of breast cancer diagnosis. However, body composition might have changed during treatments for breast cancer.31,32 We previously showed that the body composition assessed by opportunistic CT scans during a similar time period had high intra-individual reliability in general, although the TAT area on L3 images decreased by 10% while the decrease was smaller for skeletal muscle area (−1.27%).33 For patients with body composition assessed on a CT scan taken in the later stage of the treatment period, their adipose and muscle tissue areas may be underestimated, resulting in a measurement error that may have affected the associations toward null.” (page 19)
- Expand the discussion to highlight how the findings align with or diverge from existing literature, particularly concerning the roles of sarcopenia, adiposity, and immune modulation in breast cancer.
Response: The current literature on the roles of sarcopenia, adiposity, and immune modulation in breast cancer was very limited. We now provided the following statement in the Discussion as follows:
“Data on the extent to which body composition components are related to tumor immune modulation in breast cancer are very limited. Research findings on BMI and tumor-infiltrating lymphocyte (TIL) levels were inconsistent, as studies reported a positive association,19 a negative association in patients who received neoadjuvant chemotherapy,5 and no association in triple-negative breast cancer.20 Only one study investigating specific immune cells reported that BMI changes since age 18 were associated with increased CD4 and CD163 levels in breast cancer.21” (page 17)
- The inclusion of a power analysis is commendable. However, incorporating a concise summary of underpowered markers in the main text would help readers better contextualize the study’s findings.
Response: We have now included a consider summary as follows:
“Markers with low statistical power and null association in this study should not be excluded in future investigations” (page 20)
- Address minor typographical errors and ensure consistency in terminology throughout the manuscript (e.g., standardizing the use of "CTLA4" versus "CTLA-4").
Response: We have checked consistency in terminology throughout the manuscript. We used CTLA4 per HGNC Gene symbol reports.
Round 2
Reviewer 1 Report
Comments and Suggestions for Authors
Authors have satisfactorily responded to all my concerns.